# Multiscale Energy Transfers and Conversions of Kuroshio in Luzon Strait and Its Adjacent Regions

**Zhongjie He** [1,2], **Xiachuan Fu** [1,]*[] , **Yueqi Zhao** [1] **and Xuyu Jiang** [1]

[1] College of Intelligent Systems Science and Engineering, Harbin Engineering University, Harbin 150001, China; hzj@hrbeu.edu.cn (Z.H.); zhaoyqi@hrbeu.edu.cn (Y.Z.); 1053993573@hrbeu.edu.cn (X.J.)
[2] Qingdao Innovation and Development Base, Harbin Engineering University, Qingdao 266404, China
* Correspondence: fun_cent@hrbeu.edu.cn

**Abstract:** Using the local multiscale energy and vorticity analysis (MS-EVA) and based on the global high-resolution ocean reanalysis product GLORYS12V1 for 20 years, this study investigates the energy transfers and conversions of Kuroshio in the Luzon Strait and its adjacent regions through three scales, namely, the climatological scale, the seasonal scale, and the eddy scale. The results show that the inverse cascades of kinetic energy dominate the energy transfer east of Luzon (at both the eddy and seasonal scales). Kuroshio transfers the climatological kinetic energy to the eddy scale through a forward energy cascade in Luzon Strait and east of Taiwan. Because the topography of Luzon Strait and Kuroshio jointly block and limit the westward propagation of non-local eddies, the eddy energy in the South China Sea west of Luzon Strait tends to depend on local forward potential energy cascades. In these subregions, potential energy drives the accumulation of kinetic energy under the action of buoyancy conversion: interannual (seasonal) potential energy as the source of multiscale energy in the Luzon Strait (the east of Taiwan).

**Keywords:** Luzon Strait; Kuroshio; energy transfer; energy conversion; energy cascade

## 1. Introduction

As a famous western boundary current in the Western Pacific Ocean, Kuroshio evolves from the North Equatorial Current (NEC) in the offshore region east of the Philippines [1]. The current flows along the east coast of Luzon Island, the Luzon Strait and the east coast of Taiwan into the East China Sea successively, and then turns northeast along the Okinawa Trough and returns to the northwest Pacific [2,3]. During the northward flow of Kuroshio, the waters with high temperature, high salinity and high momentum are transported from the tropics to the subtropical and midlatitude regions, and it also affects the local marine ecosystem and climate, even the climate of the northern hemisphere [4–7]. Faced with these outstanding properties, the Kuroshio and its vicinity have been regarded as the focus of oceanographic research for a long period of time [8–15].

In the Luzon Strait and east of Taiwan, the Kuroshio displays significant temporal and spatial variability. Chen et al. [16] proposed that the occurrence frequency of Kuroshio intrusion into the Luzon Strait shows significant decadal variation. Previous studies on the variability of the Kuroshio in Luzon Strait have suggested a possible association between such low-frequency variations and ENSO [17–19]. In addition to ENSO, the Pacific Decadal Oscillation, i.e., PDO [20,21], and the Philippine–Taiwan Oscillation, i.e., PTO [16,22] are related to the Kuroshio variabilities in the Luzon Strait. Eddy–Kuroshio interaction (especially mesoscale eddies) is another research hotspot emerging in recent decades [3,23–26]. During the same period, considerable attention has been paid to the spatio-temporal variations of the waterway east of Taiwan. Shen et al. [27] found a relationship between ENSO and Kuroshio transport in eastern Taiwan through satellite altimetry data and mooring current meter array PCM-1. Tsui and Wu [28], however, applied the growing hierarchical self-organizing map (GHSOM) method to the study and pointed out that Kuroshio transport here

is more significantly correlated with PDO. Hsin et al. [29] studied the interannual variation characteristics of Kuroshio in east of Taiwan based on remote sensing data and identified the contribution of eddies to those interannual Kuroshio variations. Using a three-dimensional numerical model and specific local observation data, Jan et al. [30] found that a cyclonic eddy and an anticyclonic eddy produced completely different effects during their interaction with the Kuroshio, and further discovered that they are correlated to the Kuroshio intrusion of Luzon Strait. In addition, the effects of local atmospheric forcing and wind stress have also been identified [14,31]. As for the Kuroshio in northeast of Luzon Island, Cheng et al. [32] revealed that it also interacts with eddies frequently and intensely.

The multiscale variabilities of the Kuroshio drive the local momentum budget and heat exchange, and they may influence each other through complex interactions [7]. In particular, the interaction between Kuroshio and eddy has been an enduring topic [14,33–36]. By analyzing the results of the numerical model, Jia et al. [33] proposed that the instability of Kuroshio front could set the stage for the generation of eddies. Correspondingly, Yin et al. [37] found that eddies can favor (inhibit) the Kuroshio onshore intrusion by changing local potential vorticity (PV) flux, which is related to the polarity of eddies. These interactions are associated with multiscale processes (e.g., baroclinic and baroclinic instabilities and energy cascades), which can be addressed and solved by multiscale energy analysis satisfactorily [38–41]. Using the quasi-global eddy-resolving ocean general circulation model (OGCM) output from a 0.1° high resolution, Wang et al. [14] noted that baroclinic and baroclinic instabilities play different roles in several stages of Kuroshio-eddy interaction in east of Taiwan. Based on the high-resolution numerical model ROMS, Zhao et al. [34] studied the energetic process of an anticyclone eddy when Kuroshio intruded into the Luzon Strait, and found the characteristics of multistage energy cascades in the process of eddy movement. Through numerical simulation and energy analysis in the South China Sea region, Feng et al. [35] confirmed that the seasonal variation of the Kuroshio intrusion plays a key role in adjusting multiscale energy and energy budget in the northern South China Sea.

In recent decades, a number of studies have been carried out on the energetics of Kuroshio and its adjacent regions, but the time span of data is limited, and the research methods have idealized assumptions. According to Yang and Liang [6], there are at least three scale windows in the Kuroshio extension, namely, the mean flow window, the interannual-scale window and the transient eddy window, with strong energy transfers among each other. However, the study of energy transfers in the Luzon Strait and its adjacent areas from the perspective of multiscale windows is insufficient, especially based on long-term data (more than a decade). In addition, the energy sources and transfers of high-frequency eddies in the research area are still controversial. This time, we will explore the above problems by applying a multiscale analysis tool, the local multiscale energy and vorticity analysis (MS-EVA) developed by Liang and Robinson [38,42]. The analysis of the multiscale interaction modes and characteristics is expected to assist numerical simulations of this region. The rest of this paper is structured as follows. We briefly introduce the GLORYS12V1 reanalysis dataset and MS-EVA method in Section 2. In Section 3, the spectral analysis results are given as the basis of multiscale decomposition. The major results (including multiscale energy and multiscale transfer) are presented in Sections 4 and 5. Section 6 concludes this research.

## 2. Data and Methods

### 2.1. Data

Due to the limited duration and coverage of in situ observations, we use the global ocean reanalysis GLORYS12V1 of Copernicus Marine Environmental Monitoring Service (CMEMS). The reanalysis product is based on NEMO v3.1 coupled with sea ice model LIM2 with an average horizontal resolution of 1/12°. Vertically, the model has 50 depth layers with resolutions ranging from less than 1 m near the surface to 450 m near the bottom, with a maximum depth of 5902 m. GLORYS12V1 assimilates multiple types of available satellite and field data using SEEK, a singular evolutive extended Kalman filter, combined with

three-dimensional variational multivariate background error covariance matrix and a 7-day assimilation cycle [43]. Additional information about this type of data product is provided in more detail by Jean-Michel et al. [44].

Specifically, we extract daily mean temperature and salinity and three-dimensional flow fields (zonal velocity component *u* and meridional velocity *v*) for a total of 20 years from 1996 to 2015. The research area (13° N–25° N, 115° E–127° E) covers the whole Luzon Strait, part of the northwest Pacific Ocean and the northeast South China Sea. Depth layers from 0.5 to 763.3 m are selected. In order to refine the research work, according to the description in Section 1, the region is divided into four subregions: northern South China Sea (SCS; 18° N–20° N, 115° E–119° E), Luzon Strait (LS; 19° N–22° N, 119° E–122° E), the east of Taiwan (WPO; 22° N–25° N, 121° E–124° E) and the east of Luzon (LUZ; 17° N–20° N, 122° E–125° E). See Figure 1 for more details.

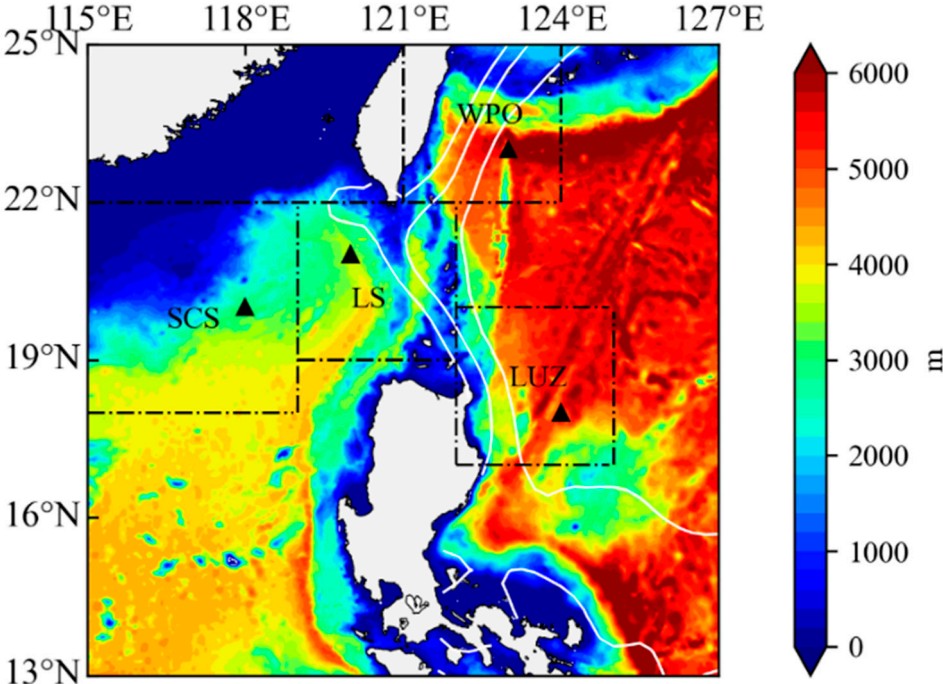

**Figure 1.** Schematic diagram of the research area and four sub-regions. They are: eastern Luzon island (LUZ), Luzon Strait (LS), eastern Taiwan-Northwest Pacific Ocean (WPO) and South China Sea (SCS). The positions of single points used for spectral analysis are represented by black triangles, which are 18° N, 124° E; 21° N, 120° E; 23° N, 123° E; 20° N, 118° E, respectively. The base map data are from ETOPO1 (unit: m) and mean dynamic topography. Here, only 110, 120 and 130 cm contour lines of MDT (white) are drawn to mark the Kuroshio axis.

### 2.2. Methods

Reynolds Mean-eddy Decomposition (MED), a technique widely used in hydrodynamic multiscale processes, can handle the interaction between the mean state of flow and eddies in stationary processes [45,46]. However, the application of this technique is difficult when the dynamic processes is non-stationary [46].

Considering the abundant multiscale ocean dynamics phenomena and strong multiscale interaction processes in our selected area, we use the multiscale energy and vorticity analysis (MS-EVA) proposed by Liang and Robinson [38]. This method is based on multiscale window transform (MWT), which was elaborated and proved in detail by Liang and Anderson [46]. MWT makes a sequence with spatio-temporal information be orthogonally decomposed into the sum of subspaces of different scales through functional transformation, and this decomposition process can maintain the local characteristics of physical processes [34]. Each subspace corresponds to a specific time scale range, also known as a scale window. According to the research needs, function space can be divided into

three scale windows: climatological window (i.e., $\varpi = 0$; for time series $u$, $\widehat{u}^{\sim 0}$), seasonal window ($\varpi = 1$; $\widehat{u}^{\sim 1}$) and eddy window ($\varpi = 2$; $\widehat{u}^{\sim 2}$). Given the length of time series $\tau$, the corresponding time ranges are $\left[2^{-j_0}\tau, \tau\right]$, $\left[2^{-j_1}\tau, 2^{-j_0}\tau\right]$, and $\left[2^{-j_2}\tau, 2^{-j_1}\tau\right]$. Here, $j_0$, $j_1$ and $j_2$ successively represent the "boundary" of the above three scale Windows and satisfy $j_0 < j_1 < j_2$ (See Section 3 for details). Within the MWT framework (See Appendix A), kinetic energy (KE, $K^\varpi$) and available potential energy (APE, $A^\varpi$) are defined as:

$$K^\varpi = \frac{1}{2}\rho_0 \widehat{\mathbf{V}}_h^{\sim\varpi} \cdot \widehat{\mathbf{V}}_h^{\sim\varpi} \tag{1}$$

$$A^\varpi = \frac{1}{2}c\left(\widehat{\rho}^{\sim\varpi}\right)^2 \tag{2}$$

where the horizontal velocity vector and density perturbation are $\mathbf{V}_h$ and $\rho$ (the density perturbation in this research is the value of density minus its background profile mean $\overline{\rho}(z)$) and $\rho_0$ represents the reference density (1025 kgm$^{-3}$). In Equation (2), $c = g^2/\left(\rho_0 N^2\right)$, where $N$ is the buoyancy frequency (i.e., Brunt-Vaisala frequency).

Further, the evolution equations of KE and APE can be derived from the Navier–Stokes equations [15,38], respectively:

$$\frac{\partial K^\varpi}{\partial t} = \Delta Q_K^\varpi + \Delta Q_P^\varpi + \Gamma_K^\varpi - b^\varpi + F_K^\varpi \tag{3}$$

$$\frac{\partial A^\varpi}{\partial t} = \Delta Q_A^\varpi + \Gamma_A^\varpi + b^\varpi + S_A^\varpi + F_A^\varpi \tag{4}$$

The physical significance and expressions of the items on the right of Equations (3) and (4) are shown in Table 1, and the derivation process of each component is referred to Liang [47].

**Table 1.** Expressions and physical meanings of items on the right in Equations (3) and (4).

| Term | Expression | Physical Meaning |
|------|-----------|------------------|
| $\Delta Q_K^\varpi$ | $-\nabla\cdot\left[\frac{1}{2}\rho_0\left(\widehat{\mathbf{V}\mathbf{V}_h}\right)^{\sim\varpi}\cdot\widehat{\mathbf{V}}_h^{\sim\varpi}\right]$ | KE advection transport on $\varpi$ |
| $\Delta Q_P^\varpi$ | $-\nabla\cdot\left(\widehat{\mathbf{V}}_h^{\sim\varpi}\widehat{P}^{\sim\varpi}\right)$ | Pressure work of KE on $\varpi$ |
| $\Gamma_K^\varpi$ | $\frac{1}{2}\rho_0\left[\left(\widehat{\mathbf{V}\mathbf{V}_h}\right)^{\sim\varpi} : \nabla\widehat{\mathbf{V}}_h^{\sim\varpi} - \nabla\cdot\left(\widehat{\mathbf{V}\mathbf{V}_h}\right)^{\sim\varpi}\cdot\widehat{\mathbf{V}}_h^{\sim\varpi}\right]$ | Canonical KE transfer to window $\varpi$ [1] |
| $\Delta Q_A^\varpi$ | $-\nabla\cdot\left[\frac{1}{2}c\widehat{\rho}^{\sim\varpi}\left(\widehat{\mathbf{V}\rho}\right)^{\sim\varpi}\right]$ | APE advection transport on $\varpi$ |
| $\Gamma_A^\varpi$ | $\frac{1}{2}c\left[\left(\widehat{\mathbf{V}\rho}\right)^{\sim\varpi}\cdot\nabla\widehat{\rho}^{\sim\varpi} - \widehat{\rho}^{\sim\varpi}\nabla\cdot\left(\widehat{\mathbf{V}\rho}\right)^{\sim\varpi}\right]$ | Canonical APE transfer to window $\varpi$ |
| $S_A^\varpi$ | $\frac{1}{2}\widehat{\rho}^{\sim\varpi}\left(\widehat{w\rho}\right)^{\sim\varpi}\frac{\partial c}{\partial z}$ | APE vertical shear on $\varpi$ |
| $b^\varpi$ | $g\widehat{\rho}^{\sim\varpi}\widehat{w}^{\sim\varpi}$ | Buoyancy conversion on $\varpi$ |
| $F_K^\varpi\left(F_A^\varpi\right)$ | —— | Dissipation terms on $\varpi$ |

[1] In canonical KE transfers, the colon operator is defined as (AB):(CD) = (A·C) (B·D), where AB and CD are both dyadic products.

According to the further analysis of interaction terms by Liang and Robinson [38], a complete energy diagram based on three-window decomposition can be obtained, as shown in Figure 2. For convenience, here we use the superscript $0\rightarrow2$ to represent the transfer from window $\varpi = 0$ to window $\varpi = 2$. For example, the transfer of KE (APE) from $\varpi = 0$ to $\varpi = 2$ is denoted as $\Gamma_K^{0\rightarrow2}(\Gamma_A^{0\rightarrow2})$. The energy transfer between other scale windows can be similarly deduced (See Appendix B).

It should be noted that since we focus on the energy transfer (conversion) between the three scale windows, the energy diagram presented in this paper only preserves the transfer between KEs (APEs) and KE–APE. In fact, these multiscale energy transfer (conversion) processes should be balanced with advection, pressure work, wind stress and buoyancy [48].

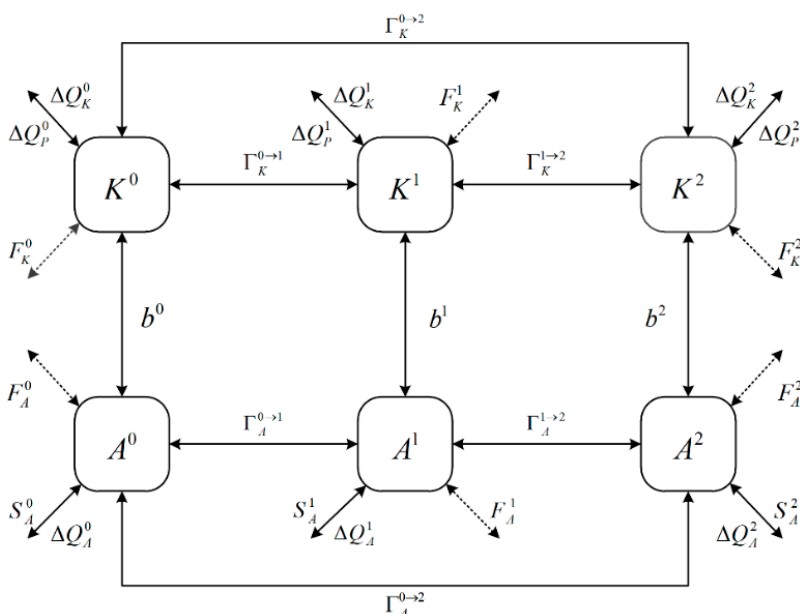

**Figure 2.** The energy diagram based on three-window decomposition. The solid arrows in both directions represent the energy transport and transfer terms, while the dashed arrows represent the energy dissipation terms.

## 3. Spectral Analysis and Scale Decomposition

### 3.1. Spectral Analysis

In the preliminary stage of this study, frequency spectrum analysis helps us to figure out the dominant components of each variable in our research area [6,48], and then specify the cutoff periods of scale decomposition (i.e., the scale window "boundaries" $j_0$, $j_1$ and $j_2$ mentioned in Section 2.2). We note the contribution of temperature to density and even potential energy, and only perform spectral analysis on three-dimensional temperature. Figure 3 shows the frequency spectra of potential temperature (*T*) at different depths of sampling points in the four subregions. The spectrum here contains no constant term (i.e., the time average of temperature).

The potential temperature spectra have substantial multiscale variations at multiple depth layers in all subregions. In general, the frequency spectra are dominated by the annual period. Above 100 m, in particular, the amplitude of the annual period is much higher than that of the other components (Figure 3a,b). At the levels below 200 m, interannual signals are also prominent, mainly concentrated near 2.5-year component (Figure 3d–f). At the same time, seasonal and interseasonal signals (with a period less than one year) in the four subregions are relatively weak, and their contributions to the time series are finite.

In order to further illustrate the vertical distribution characteristics of the above main signal components (interannual, annual, interseasonal and eddy), we provide their vertical profiles, as shown in Figure 4. Near the surface, the interannual component is weak. As the depth increases, it surges in LS and SCS, exceeding other subregions (about 100 m), while WPO contains abundant interannual signal below 400 m (Figure 4a). In Figure 4b, the annual signals of all subregions are dominant in the surface and mixed layers, but decay sharply between 25 and 100 m, even weaker than interannual component (e.g., LUZ and WPO). The amplitude of semi-annual component also exceeds that of interannual component (Figure 4c), indicating that seasonal variation is the main factor affecting the upper temperature in the study area. The high-frequency signals are roughly distributed in the thermocline below 100 m. WPO, in particular, presents a double-peak structure, located at 110 m and 450 m, respectively (Figure 4d).

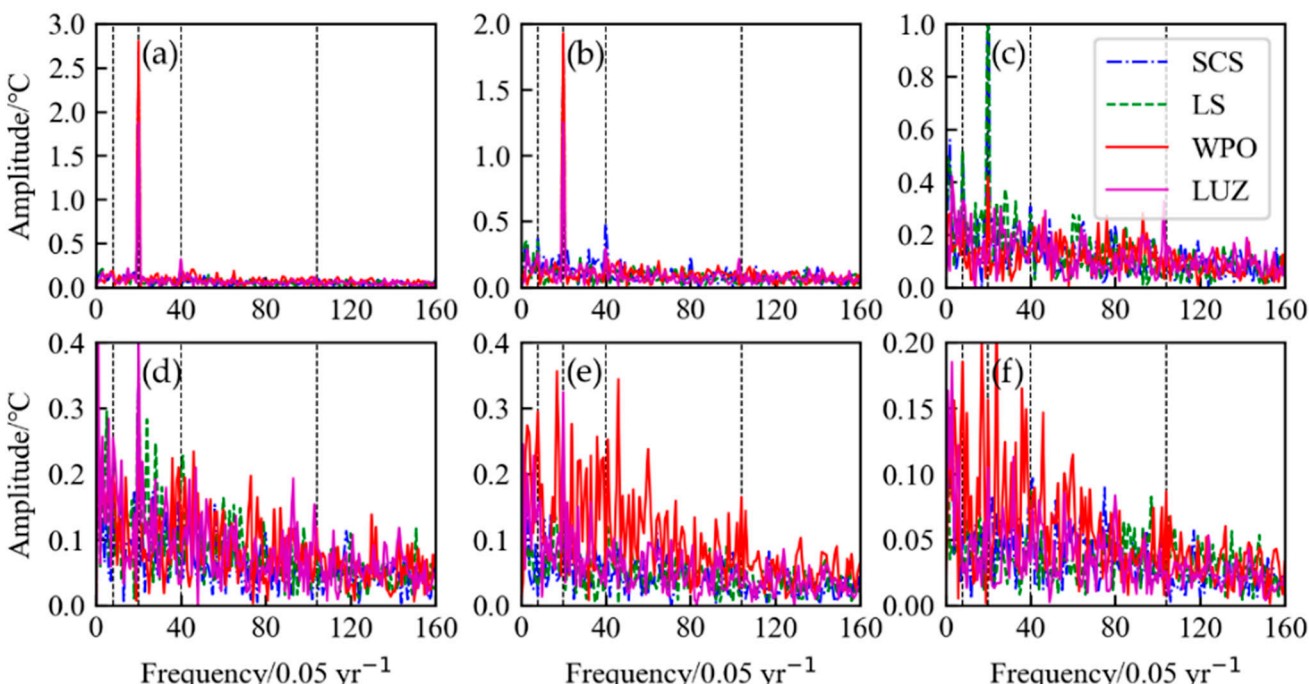

**Figure 3.** Potential temperature frequency spectrum of four subregions at different depths (see Figure 1. for the location of subregions and sampling points). The black dotted lines (from left to right) in each subgraph indicate periods of 2.5, 1, 0.5, and 0.2 year, respectively. (**a**) 25 m; (**b**) 56 m; (**c**) 110 m; (**d**) 222 m; (**e**) 454 m; (**f**) 644 m.

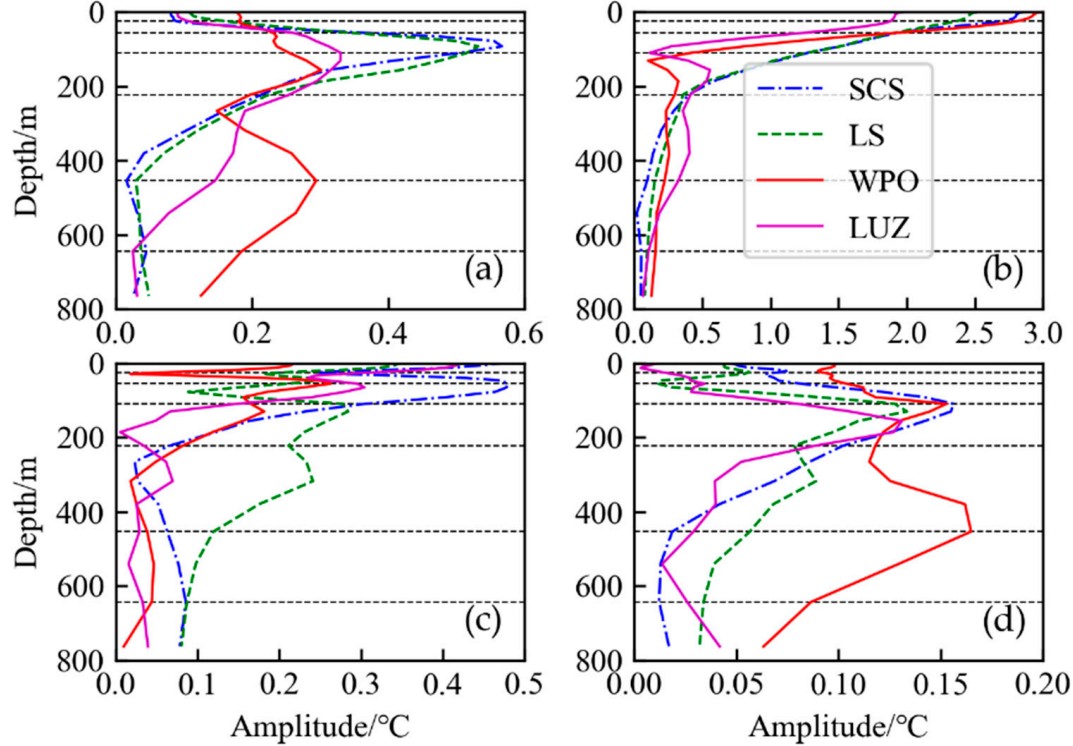

**Figure 4.** Vertical profiles of each of the four selected signal components (see Figure 3 for details). The black dotted lines (from top to bottom) in each subgraph indicate water depths of 25, 56, 110, 222, 454 and 644 m, respectively. (**a**) 2.5-year; (**b**) 1-year; (**c**) 0.5-year; (**d**) 0.2-year cycles, respectively.

### 3.2. Scale Decomposition

Based on the spectral analysis results shown in Section 3.1, we set cutoff periods $j_0$ and $j_1$ as 448 days and 112 days, respectively (i.e., $7 \times 2^6$ days and $7 \times 2^4$ days, and the time step of GLORYS12V1 data input as 7 days) to separate high-frequency signals from seasonal fluctuations or even from components with lower frequencies. Firstly, the cutoff periods mentioned above are selected because of the differences in spatial patterns between multiscale components. Secondly, there are intense seasonal variations in these subregions, which need to be classified as an independent scale window. Finally, considering the limited lifespans of eddies in the Luzon Strait and its adjacent areas [35,49,50], interference of information at other scales to the decomposition process should be reduced. In summary, MWT is used to decompose the original data into three scale windows: climatological-scale window (period > 448 days), seasonal-scale window (period between 112 days and 448 days) and eddy window (period < 112 days). Note that the term "climatology" in this study describes both the climatological seasonal cycle and interannual variability, although the interannual variability is verified to be weak compared to the seasonal cycle.

To examine the actual effect of scale decomposition, Figure 5 exhibits the original data of sea surface height field (Figure 5a,e) and the decomposition results of the three windows. There is no hiding the fact that the climatological field of SSH has a distinct height difference between the two flanks of the Kuroshio axis (Figure 5b,f). The seasonal SSH field reveals an opposite variation pattern with a boundary of 19° N (Figure 5c,g). As for the eddy components obtained by decomposition (see Figure 5d,h), the distribution, shape and duration of eddies are similar to those of previous studies [24,51]. Compared with the decomposition results of other scales, the spatial distribution of the SSH field at the eddy scale is irregular. The above properties prove the rationality of our decomposition and configuration of multiscale windows.

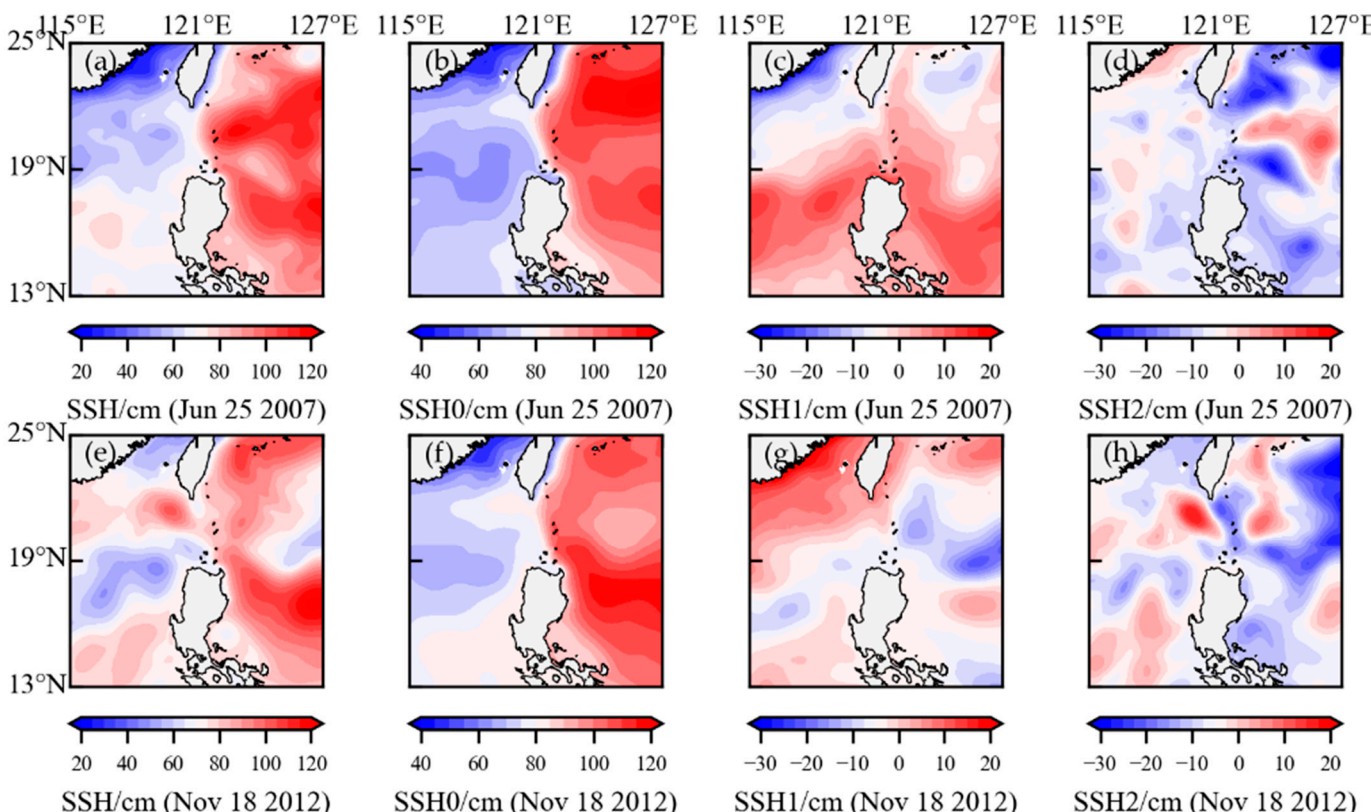

**Figure 5.** SSH snapshots (cm) on Jun 25, 2007 (**top**) and Nov 18, 2006 (**bottom**), with (**a**,**e**) the original SSH field; (**b**,**f**) the MWT climatological scale; (**c**,**g**) the seasonal scale; (**d**,**h**) the eddy scale.

## 4. Multiscale Energy

In this section, we explore the multi-year mean spatial structures of energy in three scale windows. Here, the kinetic energy (KE) of three scale windows can be computed by using Equation (1), namely, the climatological-scale KE (CKE, $K^0$), the seasonal-scale KE (SKE, $K^1$) and the eddy-scale KE (EKE, $K^2$). Similarly, three scales of APE can be obtained by Equation (2), namely, climatological-scale APE (CAPE, $A^0$), seasonal-scale APE (SAPE, $A^1$) and eddy-scale APE (EAPE, $A^2$).

Figure 6a–c shows the horizontal distributions of vertically averaged KE components (<266 m). In general, the CKE component is distributed along the Kuroshio axis principally (Figure 6a). Compared with CKE, the magnitude of SKE is much lower and mainly occurs east of Taiwan or even south of Ryukyu Islands (Figure 6b). As an indicator of eddy activity, EKE appears local maxima in the subtropical area of the northwest Pacific Ocean (east of Taiwan), and eddies are also active in a broad region from the western Luzon Strait to the northern part of the South China Sea (Figure 6c), while the relatively weak EKE occupies the west of Luzon Island. The properties of these EKE horizontal distributions have been demonstrated in previous studies [49,50]. However, we observe that the low eddy kinetic energy appears in the eastern Luzon Strait (roughly located near the island chain between 19° N and 22° N, 121° E and 122° E), which produced an evident discrepancy of magnitude from EKE on both sides of the strait.

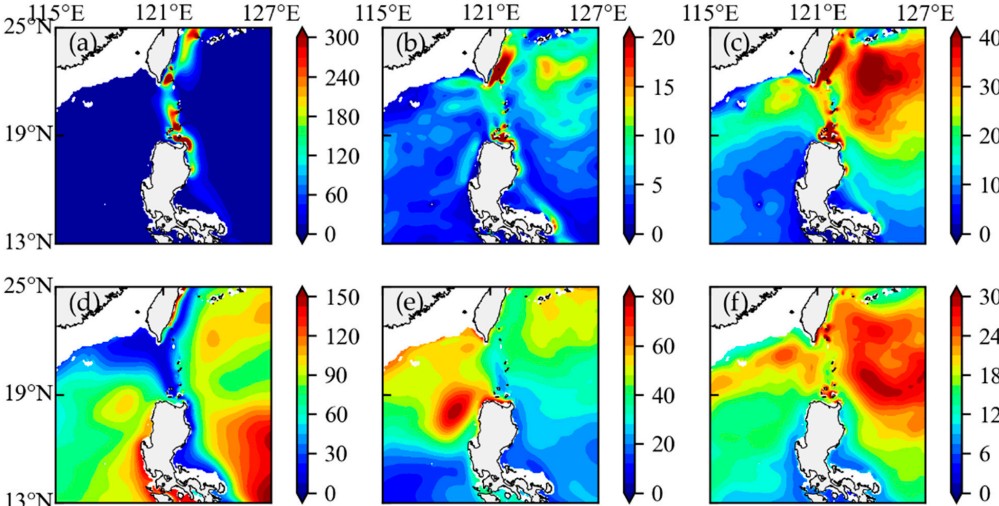

**Figure 6.** Temporally (1996–2015) and vertically (upper 266 m) averaged multiscale KE and APE components (Jm$^{-3}$). (**a**) CKE; (**b**) SKE; (**c**) EKE; (**d**) CAPE; (**e**) SAPE; (**f**) EAPE.

Figure 6d–f shows the horizontal distributions of multiscale APE components. Note that CAPE on the main axis is lower than on both sides (Figure 6d). This phenomenon is also prevalent in previous studies on Kuroshio multiscale energy [6,7,34], indicating that the storage form of climatological-scale energy on the Kuroshio axis is completely opposite to its two flanks. The magnitude of SAPE is generally higher than that of SKE (Figure 6e), indicating that energy at seasonal scale mainly exists in the form of potential energy, especially in the east of Taiwan and the northern part of the South China Sea. EAPE and EKE are not only consistent in horizontal pattern, but also have the smallest difference of magnitude among all scale windows (Figure 6f). What is noteworthy is that the magnitude of EAPE also declined sharply east of the Luzon Strait. Combined with the results of Figure 6c, it means that only a few of the eddies originating in the northwest Pacific Ocean can enter the South China Sea due to the blocking of Kuroshio current and strait topography [1,24,52]. Therefore, the direct effect of such non-local eddies on the eddy-scale energy in the northern South China Sea is extremely limited. It is necessary to explore its energy sources and transfer pathways in the following sections.

The vertical average horizontal structures of each component below the subsurface layer are shown in Figure 7. Each component of KE basically maintains the distribution trend of Figure 6a–c, but the magnitude decreased significantly (Figure 7a–c). Meanwhile, the eastern Luzon Strait still has a significant barrier effect on EKE (Figure 7c). APE components, however, are concentrated in the northwest Pacific (Figure 7d–f), and locally even exceeded the vertical mean value of CAPE above 266 m (Figure 7d).

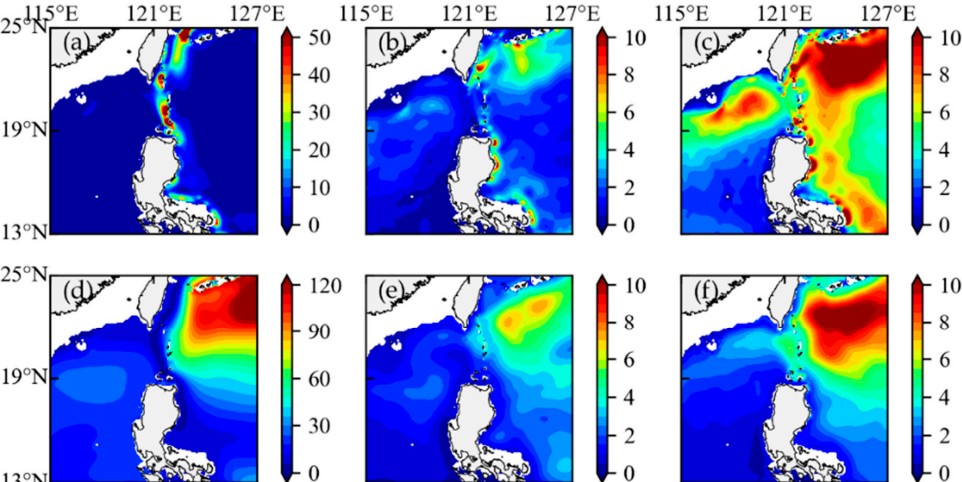

**Figure 7.** Temporally (1996–2015) and vertically (lower 266 m) averaged multiscale KE and APE components ($Jm^{-3}$). (**a**) CKE; (**b**) SKE; (**c**) EKE; (**d**) CAPE; (**e**) SAPE; (**f**) EAPE.

At the end of this section, the vertical structures of six energy components are displayed in Figure 8. Overall, the kinetic energy at each scale decays rapidly with increasing depth (Figure 8a). Of these, all KE components of WPO are the most prominent (Figure 8a–c). Different from KE, complex vertical structures appear in APE (Figure 8d,f): CAPE and EAPE decline promptly from sea surface to 50 m; between 50 and 200 m, CAPE and EAPE recover to a certain extent; below 200 m, APEs decrease again with increasing depth. During this process, CAPE is most pronounced near 200 m in the LUZ subregion and abundant between 400 m and 600 m in the WPO subregion (Figure 8d). As for the SAPE component, its surface layer (above 50 m) behaves as the largest reservoir among the three APE components (Figure 8e).

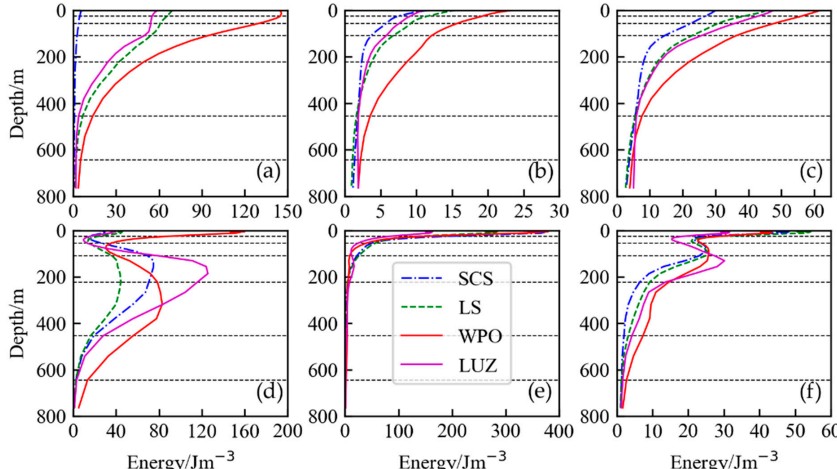

**Figure 8.** Profiles of temporally and horizontally averaged multiscale KE and APE components in four subregions (see Figure 1 for locations). The black dotted lines (from top to bottom) in each subgraph indicate depths of 25, 56, 110, 222, 454 and 644 m, respectively. (**a**) CKE; (**b**) SKE; (**c**) EKE; (**d**) CAPE; (**e**) SAPE; (**f**) EAPE.

## 5. Multiscale Interactions

In this section, the time-mean spatial distribution of multiscale interactions in the Luzon Strait and its adjacent areas is explored by computing the energy transfer among the climatological, seasonal and eddy windows. As introduced in Section 2.2, according to the canonical transfer theory of Liang [47], we can obtain the local energy transfers of climatological-eddy, seasonal-eddy and climatological-seasonal, respectively. Figure 9 shows the vertical average horizontal distribution of each component (upper 266 m).

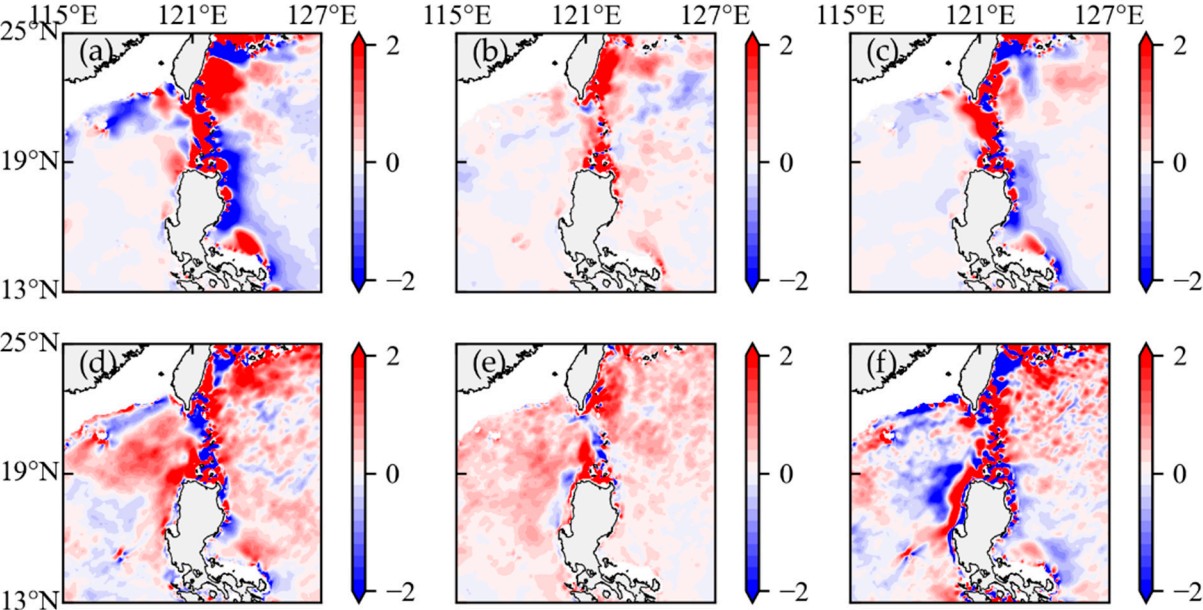

**Figure 9.** Temporally (1996–2015) and vertically (upper 266 m) averaged canonical transfers ($10^{-5}$ Wm$^{-3}$). (**a**) $\Gamma_K^{0 \to 2}$; (**b**) $\Gamma_K^{1 \to 2}$; (**c**) $\Gamma_K^{0 \to 1}$; (**d**) $\Gamma_A^{0 \to 2}$; (**e**) $\Gamma_A^{1 \to 2}$; (**f**) $\Gamma_A^{0 \to 1}$.

### 5.1. Climatological-Eddy Interaction

Figure 9a,d display the horizontal distributions of energy transfers between the climatological scale and the eddy scale. $\Gamma_K^{0 \to 2}$ occupies the axis of Kuroshio, and the forward kinetic energy cascade is dominant in Luzon Strait and eastern Taiwan (CKE→EKE, Figure 9a). In the northeast of Luzon island and the north of the South China Sea, large-scale CKE obtains energy from EKE through inverse cascade (EKE→CKE). Different from $\Gamma_K^{0 \to 2}$, the coverage of $\Gamma_A^{0 \to 2} > 0$ is larger, indicating that the cascade of potential energy is mainly positive in the whole research area, namely CAPE→EAPE (Figure 9d). The intensity of energy transfer decreases below the subsurface, but the above characteristics still exist (Figure 10a,d).

To further explore the vertical structures of energy transfers, the sections of all subregions are shown in Figure 11. Corresponding to Figure 9a,d, the WPO subregion in east of Taiwan has a strong forward energy cascade (i.e., $\Gamma_K^{0 \to 2} > 0$ and $\Gamma_A^{0 \to 2} > 0$), but the magnitude of $\Gamma_K^{0 \to 2}$ is much larger than that of $\Gamma_A^{0 \to 2}$ (Figure 11a solid and dotted red lines). It indicates that the energy transfer between climatological and eddy is dominated by kinetic energy cascade. This high intensity kinetic energy cascade continues even near the 400 m layer, which is the deepest of all regions. The situation that kinetic energy cascade exceeds potential energy cascade also exists in the Luzon Strait (LS). Through the solid purple line in Figure 11a, we also find that the inverse kinetic energy cascade in the LUZ subregion (Figure 9a) roughly occupies in the range from 50 to 150 m layers.

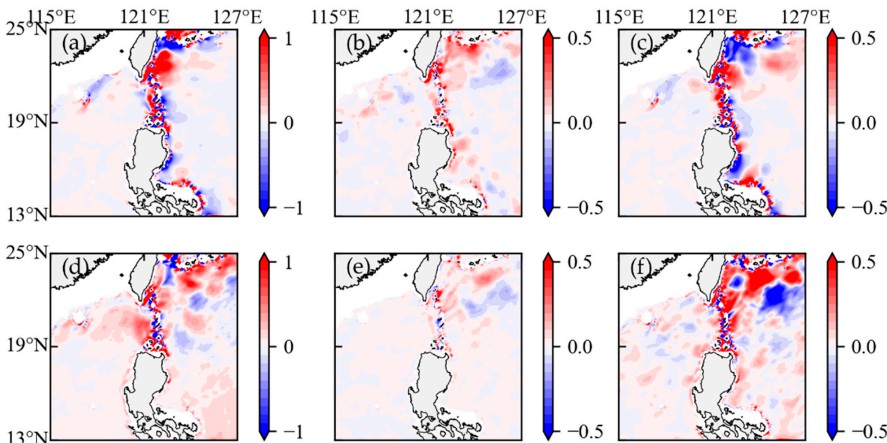

**Figure 10.** Temporally (1996–2015) and vertically (lower 266 m) averaged canonical transfers ($10^{-5}$ Wm$^{-3}$). (**a**) $\Gamma_K^{0\to2}$; (**b**) $\Gamma_K^{1\to2}$; (**c**) $\Gamma_K^{0\to1}$; (**d**) $\Gamma_A^{0\to2}$; (**e**) $\Gamma_A^{1\to2}$; (**f**) $\Gamma_A^{0\to1}$.

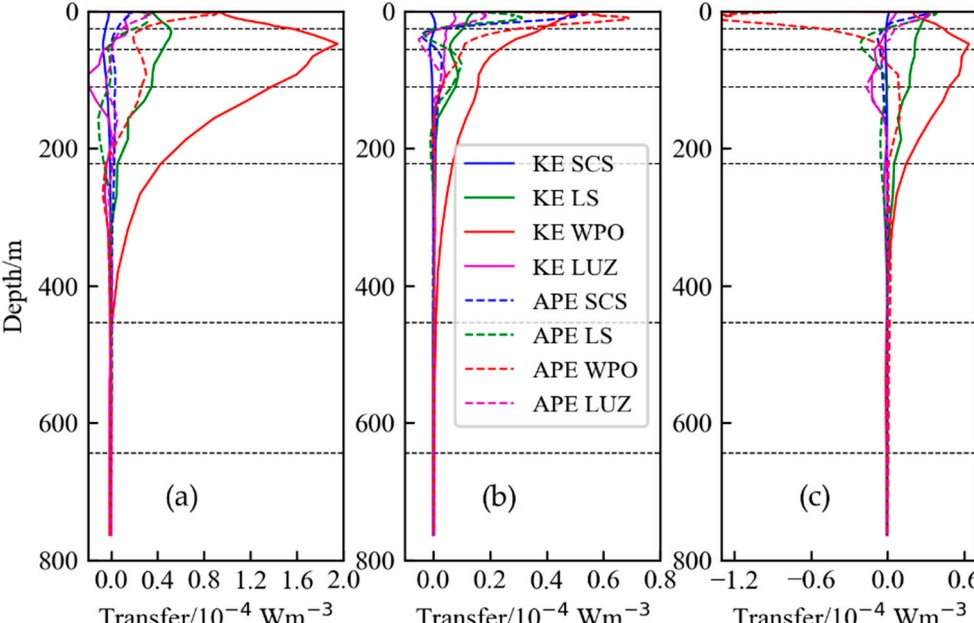

**Figure 11.** Profiles of temporally and horizontally averaged canonical transfers in four subregions (see Figure 1 for locations). The black dotted lines (from top to bottom) in each subgraph indicate depths of 25, 56, 110, 222, 454 and 644 m, respectively. (**a**) $\Gamma_K^{0\to2}$ and $\Gamma_A^{0\to2}$; (**b**) $\Gamma_K^{1\to2}$ and $\Gamma_A^{1\to2}$; (**c**) $\Gamma_K^{0\to1}$ and $\Gamma_A^{0\to1}$.

### 5.2. Seasonal-Eddy Interaction

It is not difficult to observe from Figure 9b,e that $\Gamma_K^{1\to2} > 0$ and $\Gamma_A^{1\to2} > 0$ are densely distribute in the Luzon Strait and the east of Taiwan, which proves that there is a forward cascade of seasonal scale energy in these subregions. This form of energy transfer continues in the layers below the subsurface (Figure 10b,e). From the vertical section, we can also determine the general trend of energy cascade from seasonal scale window to smaller scale ($\Gamma_K^{1\to2} > 0$ and $\Gamma_A^{1\to2} > 0$, Figure 11b). Therefore, we recognize that the forward energy cascade of the seasonal scale window also play an energetic role on the high-frequency variation of the axis (such as LS and WPO) and even the generation and development of eddies (SCS subregion). In the vertical structure, the energy transfer basically occurs above 200 m, while the $\Gamma_K^{1\to2}$ of WPO extends to near 400 m (solid red line in Figure 11b, SKE→EKE).

### 5.3. Climatological-Seasonal Interaction

On the basis of Sections 5.1 and 5.2, we continue our study of climatological and seasonal energy transfer in the Luzon Strait and its adjacent regions. The principal axis of Kuroshio is still the core area for energy transfer between the two scales. $\Gamma_K^{0\to1}$ is the most prominent in Luzon Strait, where the forward kinetic energy cascade promotes the transfer of CKE to SKE (Figure 9c). Both forward and inverse KE cascades are accumulated in eastern Taiwan. In the northeast of Luzon, the KE transfer is predominantly negative, indicating that large-scale windows absorb kinetic energy from seasonal scales. Different from kinetic energy, the transfer direction of APE in east of Taiwan is chiefly from seasonal to climatological scale (Figure 9f, $\Gamma_A^{0\to1} < 0$, SAPE→CAPE). For deeper vertical means (Figure 10c), $\Gamma_K^{0\to1}$ appears along the Kuroshio axis as in Figure 9c, while in the Northwest Pacific region, the magnitude of the APE cascade is not negligible (Figure 10f).

The profiles of the mean of the four subregions exhibit that the $\Gamma_K^{0\to1}$ of WPO remains positive (solid red line and dotted red line in Figure 11c). Thus, it can be confirmed that the WPO subregion is dominated by the forward cascade of kinetic energy, namely CKE→SKE. In contrast to WPO, the value of $\Gamma_K^{0\to1}$ in LUZ subregion is negative below 50 m (Figure 11c, solid purple line dotted purple line), suggesting that SKE in this region is affected by a inverse energy cascade and energy transfers to the interdecadal scale window. For LS, the transfer directions of kinetic energy (KE) and potential energy (APE) below 25 m are opposite, that is, CKE→SKE and SAPE→CAPE, respectively (solid green line and dotted green line in Figure 11c).

### 5.4. Buoyancy Conversion

Reviewing the introduction of MS-EVA in Section 2.2, the conversion between kinetic and potential energy at the same scale also exists (Table 1). Notice here, b > 0 represents the direction of energy conversion from KE to APE in this scale window. The magnitude of $b^0$ at the climatological scale is higher on the Kuroshio axis than on both sides (Figure 12a,d). For Kuroshio to the east of Taiwan, its direction of conversion ($b^0$) is controlled by CAPE→CKE. The seasonal scale window conversion accumulated in the Luzon Strait and east of Taiwan (Figure 12b,e) with limited intensity; however, we note that $b^1$ is still negative in the east of Taiwan. As for the eddy scale, $b^2$ is negative on the whole (i.e., EAPE→EKE) in the research area, indicating that EAPE behaves as a momentous source of EKE (Figure 12c,f), and the jet axis in east of Taiwan is the most representative.

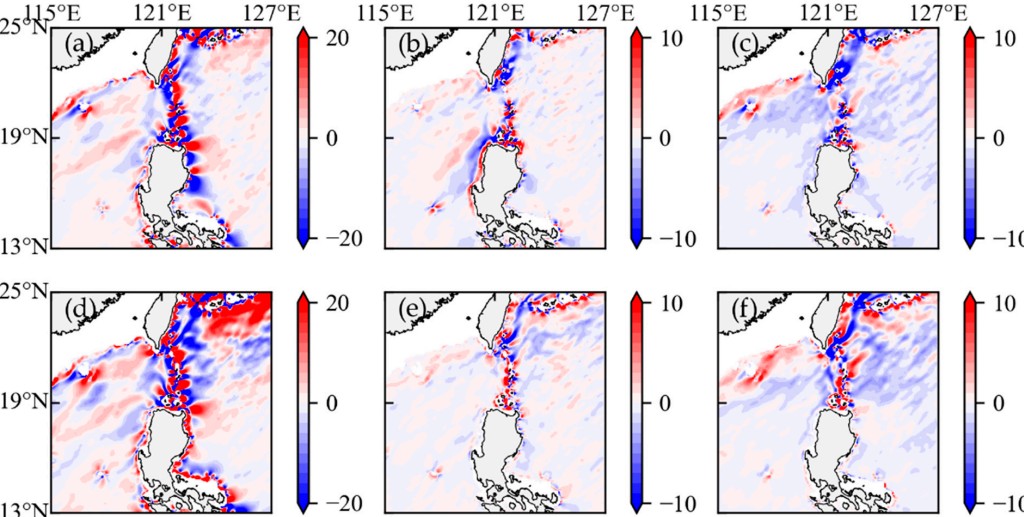

**Figure 12.** Temporally (1996–2015) and vertically averaged buoyancy conversions ($10^{-5}$ Wm$^{-3}$). (**a**) $b^0$ upper than 266 m; (**b**) $b^1$ upper than 266 m; (**c**) $b^2$ upper than 266 m; (**d**) $b^0$ lower than 266 m; (**e**) $b^1$ lower than 266 m; (**f**) $b^2$ lower than 266 m.

Figure 13 displays the vertical distribution of the buoyancy conversions for all subregions. Compared with the cross-scale transfers, the buoyancy conversions remain active below 400 m (Figure 13a,c). The conversions are significant in LS and WPO, and the direction is APE→KE, indicating that in these subregions, the APE component in each scale window is competent to be the energy source of the corresponding KE component, thus confirming our findings in Figure 12.

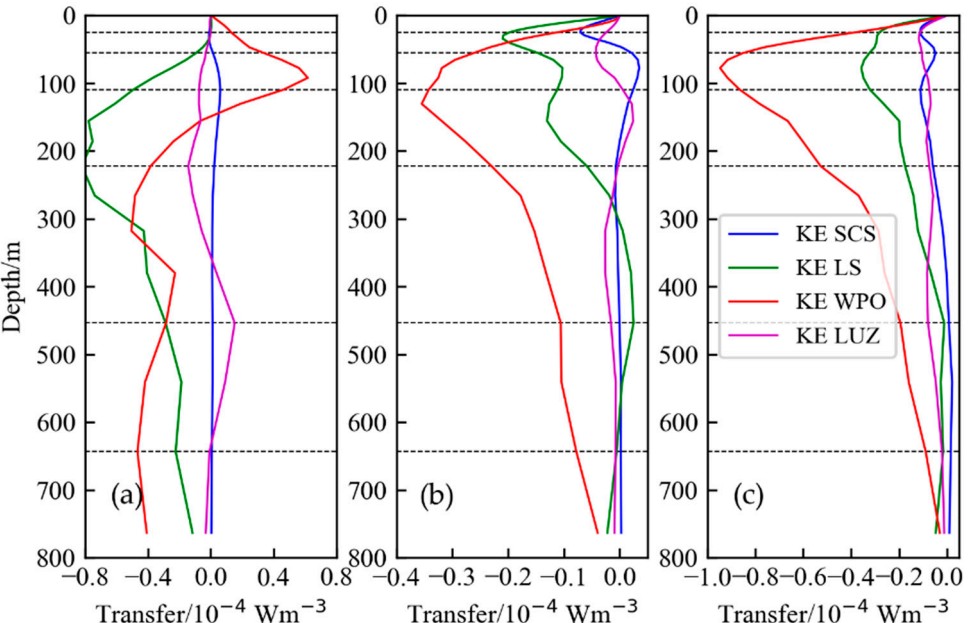

**Figure 13.** Profiles of temporally and horizontally averaged buoyancy conversion in four subregions (see Figure 1 for locations). The black dotted lines (from top to bottom) in each subgraph indicate depths of 25, 56, 110, 222, 454 and 644 m, respectively. (**a**) $b^0$; (**b**) $b^1$; (**c**) $b^2$.

### 5.5. Energy Pathway and Discussions

At the end of this section, we calculate the mean magnitude of energy transfers (conversions) and compile the energy diagrams for each region (Figure 14). Under the action of inverse kinetic energy cascade (Figures 9a,c and 14a), the climatological window absorbs KE (i.e., EKE→CKE) from the high-frequency eddies. Similarly, the seasonal scale transfers its kinetic energy to the climatological scale (SKE→CKE). On the one hand, such cascade direction indicates that the large-scale energy of the region is replenished in the process of transfer. On the other hand, it is beneficial for the spatial position of the Kuroshio axis in this region to remain stable during its interaction with the eddies [53].

Forward cascades (e.g., CKE→EKE, CAPE→EAPE) are located in LS subregion of Luzon Strait, and the transfers of kinetic energy far exceed the APE (Figure 14b). About 60% of the energy released from the climatological window directly feeds the eddy scale, which means that the small-scale phenomena in Luzon Strait are related to the forward cascade of large-scale components closely. At the same time, there is a significant buoyancy conversion at the climatological scale (Figure 12a,d). The above energy pathways prove that the CAPE is a reliable source of LS multiscale energy, and no matter what path, it can eventually sink to small-scale EKE. Among them, CAPE→CKE→EKE channel has made great contribution to the accumulation of EKE.

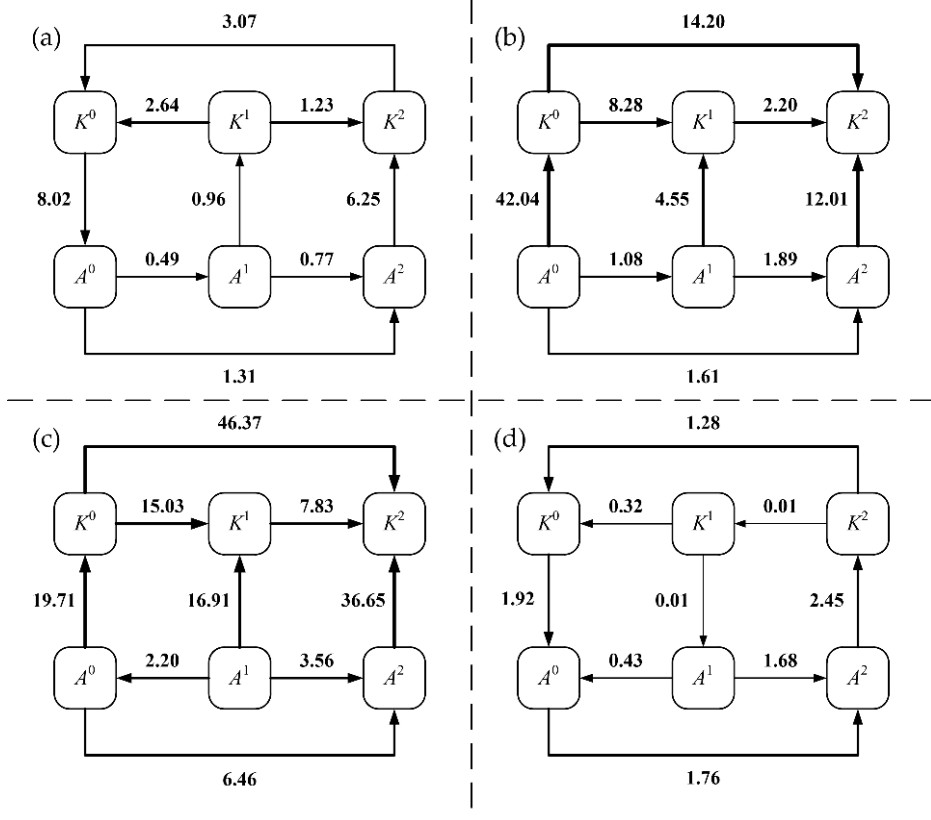

**Figure 14.** The energy diagram (refers to Figure 2; volume-averaged) for the four subregions in Figure 1. The units are in $10^{-6}$ Wm$^{-3}$. (**a**) LUZ; (**b**) LS; (**c**) WPO; (**d**) SCS.

The situation of WPO subregion east of Taiwan is analogous to that of LS, except that the local energy transfer and transformation of WPO are more intense (Figure 14c), and even extends to deeper levels below 200 m (Figures 10 and 11). In addition, SAPE becomes a source of multiscale energy. The energy transfer pattern of WPO is complicated de facto. A strong inverse energy cascade can be observed in northeastern Taiwan (Figures 9a,d and 10a,d). However, the forward energy cascades in southeast Taiwan exhibit the ability of energy transfer to smaller scales, thus affecting the stability of the Kuroshio axis. In particular, the Kuroshio appears to truncate, bend, and bifurcate during its interaction with eddies originating in the northwest Pacific Ocean [3,13].

According to the statistical results of Feng et al. and Trott et al. on the eddy in the northeast of South China Sea, the eddy activity here is also frequent [35,50]. Figure 14d reveals that the energy transfer of SCS subregion (northeastern part of South China Sea) prevailingly occurs between the climatological and eddy windows. Nevertheless, unlike other subregions, the APE cascades of SCS outperformed the KE cascades. Special attention should be paid to the restrictions of non-local eddies propagation to the South China Sea due to the topography of the Luzon Strait and Kuroshio. We conclude that local energy transfer (conversion) in SCS subregion is the main source of eddy scale energy: EAPE can receive potential energy transfer from climatological and seasonal scales simultaneously under the action of forward potential energy cascades (contribution of both is close to 1:1) and transmit to EKE through negative buoyancy conversion.

## 6. Conclusions

Using the local multiscale energy and vorticity analysis (MS-EVA) and based on the global high-resolution ocean reanalysis product GLORYS12V1 for 20 years, this study investigates the energy transfers and conversions of Kuroshio in the Luzon Strait and its adjacent regions. Including the climatological scale, the seasonal scale and the eddy

scale. Vertically, the WPO subregion in east of Taiwan (Northwest Pacific Ocean) is rich in CAPE below 400 m. As the existence form of seasonal scale energy, SAPE forms a considerable energy reservoir in the mixed layer above 50 m. EKE and EAPE of the eddy scale not only have the same horizontal distribution characteristics (for instance, in the east of Taiwan and the Luzon Strait), but also have the smallest difference of magnitude among all scale windows. In particular, by the joint blocking effect of Luzon Strait topography and Kuroshio, the direct influence of the propagation of the northwest Pacific eddies on the local eddy-scale energy in the northern South China Sea is limited.

Then, we discuss the energy transfer and conversion between different scale windows. The unique spatial pattern and energy pathways of each subregion are gradually clarified: (1) in the LUZ subregion east of Luzon, an inverse cascade between eddy kinetic energy and seasonal scale kinetic energy dominates the cross-scale energy transfer; (2) the forward KE cascade intensity of the Luzon Strait (LS) is higher than that of the APE, and the APE becomes an important source of the KE of the corresponding scale (assisted by buoyancy conversion), in which CAPE and EKE are the source and sink of local energy transfer in this region, respectively; (3) in WPO, the energy transfer (conversion) is more intense and extends down to deeper levels (below 400 m), where SAPE acts as the energy source; (4) under the condition that the propagation of non-local eddies (especially originating in the northwest Pacific Ocean) to the South China Sea is severely suppressed, the eddy energy of SCS subregion is actually dependent on the local forward potential energy cascades (i.e., CAPE→EAPE, SAPE→EAPE).

Based on the above analysis of multiscale energy and transfers, the multiscale function modes and characteristics of Luzon Strait and its adjacent areas are clarified, and it can also assist in optimizing numerical simulations in the region. Firstly, the importance of potential energy as a source of kinetic energy outlines the need to improve simulations of temperature and salinity in the research area. Secondly, for Luzon Strait and the northern part of the South China Sea, more attention should be paid to temperature and salinity correction of climatology (including interannual components). Finally, in the area east of Taiwan, seasonal and even smaller scale interseasonal fluctuations in temperature and salinity are also noteworthy when suppressing the model bias and drift. The above proposals and viewpoints need to be verified and verified through the simulation of the model. Relevant research works will be further improved in the future.

**Author Contributions:** Each author has made substantial contributions to this research. Conceptualization, Z.H. and X.F.; methodology, Z.H., X.F. and Y.Z.; data analysis, X.F., Y.Z. and X.J.; writing—original draft preparation, X.F.; writing—review and editing, Z.H. and X.F.; supervision, Z.H.; funding acquisition, Z.H. All authors have read and agreed to the published version of the manuscript.

**Funding:** The research was funded by the National Key R & D Program of China NO. 2018YFC1406202.

**Institutional Review Board Statement:** Not applicable.

**Informed Consent Statement:** Not applicable.

**Data Availability Statement:** The GLORYS12V1 reanalysis data were obtained from the Copernicus Marine Environment Monitoring Service (CMEMS, https://resources.marine.copernicus.eu, accessed on 1 November 2021). The mean dynamic data and sea level anomaly data were obtained from the Copernicus Climate Change Service (C3S, https://cds.climate.copernicus.eu/#!/home, accessed on 31 July 2021). ETOPO1 data: https://www.ngdc.noaa.gov/mgg/global/global.html. (accessed on 2 January 2022).

**Acknowledgments:** The GLORYS12V1 data were obtained from the Copernicus Marine Environment Monitoring Service (CMEMS, https://resources.marine.copernicus.eu, accessed on 1 November 2021). The mean dynamic data and sea level anomaly data were obtained from the Copernicus Climate Change Service (C3S, https://cds.climate.copernicus.eu/#!/home, accessed on 31 July 2021). ETOPO1 data: https://www.ngdc.noaa.gov/mgg/global/global.html. (accessed on 2 January 2022) This research was supported by the National Key R & D Program of China No. 2018YFC1406202. Most of the figures in this paper were created with the Basemap toolkit of Python3.8. The authors are

grateful to those researchers who provided valuable advice during the study and the writing of the manuscripts.

**Conflicts of Interest:** The authors declare no conflict of interest.

## Appendix A. Multiscale Window Transform (MWT)

The appendix shows the operation process of multiscale window transform method. For a given time series $u(t)$, its scale transform coefficient in a scale level $j$ is defined as follows:

$$\widehat{u}_n^j = \int_0^l u(t)\phi_n^j(t)dt, \quad 0 \leq j \leq j_2, \quad n = 0, 1, ..., 2^j l - 1 \tag{A1}$$

where $\phi_n^j(t)$ is a scaling and wavelet function, $n$ is the discrete time step in the sampling space and the parameter $l = 1$ or $l = 2$, corresponds, respectively, to the periodic and symmetric extension schemes [47].

Note that the definition of window boundaries in Section 2.2 is $j_0 < j_1 < j_2$, and then $u$ can be reconstructed as components of three scale windows:

$$u^{\sim 0}(t) = \sum_{n=0}^{2^{j_0}l-1} \widehat{u}_n^{j_0} \phi_n^{j_0}(t) \tag{A2}$$

$$u^{\sim 1}(t) = \sum_{n=0}^{2^{j_1}l-1} \widehat{u}_n^{j_1} \phi_n^{j_1}(t) - u^{\sim 0}(t) \tag{A3}$$

$$u^{\sim 2}(t) = u(t) - u^{\sim 0}(t) - u^{\sim 1}(t) \tag{A4}$$

Further, according to the deduction of Liang and Robinson [38], we can obtain the transform coefficient of time series $u(t)$ only within the scale window $\varpi$:

$$\widehat{u_n^{\sim \varpi}} = \int_0^l u^{\sim \varpi}(t)\phi_n^{j_2}(t)dt, \; \varpi = 0, 1, 2 \text{ and } n = 0, 1, ..., 2^j l - 1 \tag{A5}$$

## Appendix B. The Expansion of Canonical Transfer Terms

It should be noted that the energy transfer terms $\Gamma_K^\varpi$ and $\Gamma_A^\varpi$ in Equation (3), Equation (4) and Table 1 are still in cumulative forms. Therefore, in this appendix, we plan to provide their further decomposition process. This procedure is called the "interaction analysis" [38,47]. As shown in Equation (A6), all the canonical transfers can be rewritten as a combination of terms in the form:

$$\Gamma_n^\varpi = \widehat{R}_n^{\sim \varpi} \cdot \widehat{(pq)}_n^{\sim \varpi} \tag{A6}$$

To facilitate the following explanation and illustration, we here just take the particular case $\Gamma_n^2$ for example:

$$\Gamma_n^2 = \widehat{R}_n^{\sim 2} \cdot \widehat{(pq)}_n^{\sim 2} = \widehat{R}_n^{\sim 2}\left(\sum_{\varpi_1=0}^{2} p^{\sim \varpi_1} \sum_{\varpi_2=0}^{2} q^{\sim \varpi_2}\right)^{\sim 2} = \widehat{R}_n^{\sim 2}[(\widehat{p^{\sim 0}q^{\sim 0}})_n^{\sim 2} + (\widehat{p^{\sim 2}q^{\sim 0}})_n^{\sim 2} + (\widehat{p^{\sim 0}q^{\sim 2}})_n^{\sim 2}]$$
$$+ \widehat{R}_n^{\sim 2}[(\widehat{p^{\sim 1}q^{\sim 2}})_n^{\sim 2} + (\widehat{p^{\sim 1}q^{\sim 1}})_n^{\sim 2} + (\widehat{p^{\sim 2}q^{\sim 1}})_n^{\sim 2}] + \widehat{R}_n^{\sim 2}[(\widehat{p^{\sim 0}q^{\sim 1}})_n^{\sim 2} + (\widehat{p^{\sim 1}q^{\sim 0}})_n^{\sim 2}] + \widehat{R}_n^{\sim 2}(\widehat{p^{\sim 2}q^{\sim 2}})_n^{\sim 2} \tag{A7}$$

The first and second terms of Equation (A7) represent the energy transfers to scale window 2 from windows 0 ($\Gamma_n^{0 \to 2}$) and 1 ($\Gamma_n^{1 \to 2}$), respectively. The third term represents the result of the joint action of windows 0 and 1 on window 2, denoted as $\Gamma_n^{0 \oplus 1 \to 2}$, whose contribution to the overall energy transfers is negligible [48]. The final term of Equation (B2), $\Gamma_n^{2 \to 2} = \widehat{R}_n^{\sim 2}\left(\widehat{p^{\sim 2}q^{\sim 2}}\right)_n^{\sim 2}$, is the energy transfer from window 2 itself. Similarly, the expansions of canonical transfer terms of other scale windows are not difficult to derive.

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
