# Peer review of "Multiscale Energy Transfers and Conversions of Kuroshio in Luzon Strait and Its Adjacent Regions"

_jmse, doi:10.3390/jmse10070975_

Round 1
Reviewer 2 Report
This manuscript presents a detailed energy analysis of the Kuroshio region, based on high-resolution numerical simulations. The study is focused on the partition of the kinetic and potential energy among various time scales, as well as on the energy transformation between the low-frequency, seasonal and mesoscale variability. The study employs the multiscale energy and vorticity analysis, based on multiscale window transform. The method is described in the previous studies but is not explained here. I am not familiar with the technique but assume that the technical results are solid. The results are interesting, and the manuscript should be published after a major revision. The presentation is generally clear, but English must be improved before the paper can be published. I encountered many sentences whose meaning was not clear because of poor English. I tried to identify those sentences in the minor comments below, but almost certainly missed many others.
Major points:
I recommend that the multiscale analysis and window transform is described in this paper. I understand that it was done in the previous studies, but a short technical description (which can be put in an Appendix) will help a reader not familiar with this technique.
“Climatology” is used for both the low-frequency variability and the seasonal cycle in Sections 3 and 4. This is very confusing. Please use consistent terminology and define it clearly. “Climatology” is often defined as the average seasonal cycle, but then it cannot be used to describe the low-frequency variability pattern; the seasonal variability is likely to be dominated by climatology, but technically they are two different variables.
The seasonal signal is close to the mesoscale in the spectral space, and mesoscale fields can be easily contaminated by the climatological seasonal cycle. For this reason, I believe it is customary to remove the climatological seasonal cycle from data before studying the eddy-mean interactions. Can the authors explain why their study uses the full time series instead?
Minor points:
l. 16,18 and elsewhere – Use “direct cascade” instead of “positive cascade” and “inverse cascade” instead of “negative cascade” to avoid ambiguity
l.30: It is better to replace “high velocity” with “high momentum”
l.35: It is better to add “strong” or “significant” before “variability”. All currents display some sort of variability.
l.41: Please define what is meant by eddies in this study. Are we looking at mesoscale variability in time or mesoscale spatial patterns? Do eddies include submesoscale currents?
l.58: The sentence needs to be rephrased
l.62: “can favor (inhibit)”
l.77: “idealized” instead of “ideal”
ll.86-87: Rephrase “optimize the effect of numerical simulation in this region in the future research work”
l.122: “eddies” (plural). Also clarify that the “time-mean” flow is meant here.
l.124: Please explain why the Reynolds decomposition have difficulties in describing inhomogeneous processes.
l.131: A word is missing after “decomposed”. Also please clarify what “maintain the local characteristics of physical processes” means.
ll.133-136: Say that the actual time intervals will be determined later, using the spectral analysis. Otherwise, the reader is left wondering how these three classes of motions are defined.
l.159: Replace “in the following paper” with “here” or “in this paper”
ll.172-174: The sentence needs to be rephrased.
l.186: I think it is better to say that Fig.4 shows profiles of spectral power at each of the four frequencies. Caption to the figure should be clarified as well.
ll.211-212: Please clarify what this sentence means.
ll.216-218: I thought that the data are decomposed into the low-frequency, mesoscale and submesoscale components (see ll.133-138), but here the decomposition is defined very differently. The definitions must be used consistently.
ll.222-224: The climatological pattern does not look opposite to the low-frequency pattern.
ll.224-225: I do not understand what this sentence is saying.
l.243: Please rephrase or remove the “authoritative parameter to measure eddies”
l.247: The analysis cannot detect eddy movement
l.263: What is the “The EAPE plummet”?
l.306 and elsewhere: “inverse” instead of “reverse”
l.321: Remove “form”
l.367: Please remind the reader what the positive/negative values of this term mean
l.385: Rephrase “remains of a certain intensity”, it is not clear what is meant.
ll.397-398: I do not understand what this sentence is saying.
ll.417-418: How can we see that the energy transformation is due to variability in the position of the Kuroshio axis?
ll.432-438: The English in this part of the paragraph needs to be improved.
l.449: Please clarify what “the APE components are the pivotal source of the corresponding KE components” means.
ll.452-454: How do we know that the low energy in the SCS region is explained by the blocking of propagating eddies? It does not directly follow from the analysis.
The last paragraph of the Conclusions is very confusing. I do not understand how the authors arrived at the conclusions (1) to (3) and why these points can help “to optimize the numerical simulation of this area”. Please clarify or remove the entire paragraph.
Round 2
Reviewer 1 Report
I would like to thank the authors for addressing my comments. This paper can be accepted without further revisions.
Reviewer 2 Report
The authors addressed all my previous concerns, and I recommend publication after a minor revision. Specific points still requiring author attention are listed below:
ll.90-92: “On the basis of clarifying the multiscale interaction modes and characteristics of the study area-91 Then, it is expected to provide useful reference and support for us to optimize the numerical model simulation of this region” does not read well. Please rephrase to something like
“The analysis of the multiscale interaction modes and characteristics is expected to assist numerical simulations of this region”
My original comment on l.24: I was asking to explain why Reynolds decomposition cannot properly describe inhomogeneous processes. The MED is based on time averaging and can easily distinguish one location from another. Please provide a brief explanation in the paper or remove “inhomogeneous”.
l.139: a word is still missing after “decomposed”.
l.147: remove “for principles”
l.163: remove “for calculation principle”
l.187: add “the” before the “annual period”
ll.230-233: This paragraph needs rephrasing and shortening: “It is worth emphasizing that the term "climatology" as an epithet for the large-scale window in this study is not unrelated to the fact that the constant term in the temperature spectrum far exceeds any interannual component. Here we tend to define the mean state of the oceans over a period of time as climatology.”
My suggestion (assuming it reflects the intended meaning):
“Note that the term “climatology” in this study describes both the climatological seasonal cycle and interannual variability, although the interannual variability is verified to be weak compared to the seasonal cycle”
ll.500-501: I would change “and it is also of reference significance for us to optimize the numerical simulation” to “and it can also assist in optimizing numerical simulations in the region”.
ll.502-506: I assume that this sentence is saying that the importance of potential energy as a source of kinetic energy outlines the need to improve simulations of temperature and salinity in the region. Please rephrase the sentence to make it short and clear.
l.509: What is “thermohaline correction”? Does it mean “temperature and salinity” or “thermocline”? I would talk about stratification for better clarity.
l.558: “The first term represents…”
